# Effects of BPA Exposure and Recovery on the Expression of Genes Involved in the Hepatic Lipid Metabolism in Male Mice

**DOI:** 10.3390/toxics11090775

**Published:** 2023-09-12

**Authors:** Changqing Li, Nan Shen, Shaohua Yang, Hui-Li Wang

**Affiliations:** College of Food and Biological Engineering, Hefei University of Technology, Hefei 230009, Chinasn99hfut@163.com (N.S.)

**Keywords:** BPA, lipid accumulation, *APOD*, lipid homeostasis

## Abstract

Exposure to Bisphenol A (BPA) has led to an increased risk of obesity and nonalcoholic fatty liver diseases (NAFLDs). However, it is as yet unclear if the damage caused by BPA is able to be repaired sufficiently after exposure has ceased. Therefore, this project aims to investigate the effects of BPA on the hepatic lipid metabolism function and its potential mechanisms in mice by comparing the BPA exposure model and the BPA exposure + cessation of drug treatment model. Herein, the male C57BL/6 mice were exposed in the dose of 50 μg/kg/day and 500 μg/kg/day BPA for 8 weeks, and then transferred to a standard chow diet for another 8 weeks to recover. Based on our previous RNA-seq study, we examined the expression patterns of some key genes. The results showed that the mice exposed to BPA manifested NAFLD features. Importantly, we also found that there was a significant expression reversion for *SCD1*, *APOD*, *ANGPT4, PPARβ, LPL* and *G0S2* between the exposure and recovery groups, especially for *SCD1* and *APOD* (*p* < 0.01). Notably, BPA could significantly decrease the level of *APOD* protein (*p* < 0.01) whereas there was an extremely significant increase after the exposure ceased. Meanwhile, *APOD* over-expression suppressed TG accumulation in the AML12 cells. In conclusion, the damage caused by BPA is able to be repaired by the upregulation of *APOD* and exposure to BPA should be carefully examined in chronic liver metabolic disorders or diseases.

## 1. Introduction

Nonalcoholic fatty liver disease (NAFLD) is the most common liver disease and the incidence of NAFLD has sharply increased since the 1990s. It has now become a major public health concern worldwide [1]. A significant proportion of patients with NAFLD may progress to nonalcoholic steatohepatitis (NASH), which is associated with disease progression, such as fibrosis, cirrhosis, or hepatocellular carcinoma [2]. The causes of epidemic NAFLD still remain unclear and it is evident that environmental factors also play an important role in NAFLD [3]. Environmental exposures, including but not limited to insecticides, pesticides, and polychlorinated biphenyls, can increase the risk of developing NAFLD and the reason for this is that these exposures are potential fat metabolic modifiers in the liver [2].

Bisphenol A (BPA) is one of the well-known endocrine-disrupting chemicals (EDCs) [4] and it is now accepted as a factor contributing to the increasing incidence of obesity and metabolic diseases, including NAFLD, insulin resistance, type 2 diabetes, and dyslipidemia [5,6]. Several studies have indicated that metabolic disorders were observed in later life when they were exposed to BPA during the critical period of development [7,8]. Perinatal exposure to BPA leads to disruption of pathways related to adipogenesis and results in increased fat mass and body weight [9,10]. Consistent with these findings, we also observed that an early-life BPA exposure resulted in a higher body weight and fat percentage, a greater fat mass of white adipocytes and displayed a NAFLD-like phenotype in male C57BL/6 mice in our previous study [11]. Therefore, BPA could be considered as an important risk factor related to the progression of obesity and NAFLD.

Until now, the mechanism of BPA-mediated adipogenesis and NAFLD has not been thoroughly understood. Furthermore, it is as yet unclear if the damage caused by BPA is able to be repaired sufficiently for a return to normal levels. Therefore, we further investigated the changes in a series of key genes involved in TG and lipid metabolism detected by our previous RNA-seq analysis [11] during BPA exposure and post-exposure recovery. This provides a favorable basis for the potential mechanism that BPA exposure leads to metabolic diseases and can also be applied to an understanding of more general mechanisms contributing to hepatic steatosis. Therefore, the present study first investigated the effects of drug treatment and drug withdrawal on metabolic phenotypes in mice; Second, the effects of BPA on the expression of genes and proteins related to lipid metabolism in mouse livers under different treatment conditions were determined by Q-PCR and Western blot (WB); Finally, molecular interference is used to validate the function of the relevant molecules in vitro. The aim is to identify potential molecular therapeutic targets for BPA-induced hepatic lipid metabolism disorders.

## 2. Materials and Methods

### 2.1. Chemicals and Materials

Bisphenol A (purity ≥ 99%), ethanol and dimethyl sulfoxide (DMSO) were purchased from Sigma-Aldrich (Sigma, St. Louis, MO, USA). Triglyceride (TG) quantification, oil red O staining, liver TG and serum TG, total cholesterol (TCHO), high-density lipoprotein cholesterol (HDL-C) and low-density lipoprotein cholesterol (LDL-C) kits were supplied by the Jiancheng Bioengineering Institute (Nanjing, Nanjing, China). Trizol isolation kits were purchased from Invitrogen (Carlsbad, CA, USA). The total proteins lysis buffer, PVDF membrane and a BCA protein detection kit were provided by TIANGEN (Beijing, China). Primary polyclonal antibody APOD (ab236868, Abcam, Cambridge, MA, USA, Dilution ratio: 1/1000) and SCD1 (ab236868, Abcam, Cambridge, MA, USA) secondary antibody were purchased from Abcam (ab6721, Cambridge, MA, USA, Dilution ratio:1/1000). The other analytical reagents were sourced from the Servicebio Technology Co., Ltd. (Wuhan, Hubei, China), if not specified.

### 2.2. BPA Exposure and Recovery

Animal care and management were approved by the Animal Care and Use Committee of the Hefei University of Technology (approval number HFUT20210413002). Adult (2-month-old) male and female C57BL/6 mice were purchased from the Charles River. Mice were maintained under a constant temperature of 25 ± 2 °C and relative humidity of 60 ± 5% under a 12 h light/dark cycle. As perinatal mice cannot be administered by gavage, therefore after childbirth and during breastfeeding the mother mice were exposed to DMSO and BPA with doses of 50 or 500 μg/kg BW/day separately by gavage. Subsequently, all 3-week-old mice were randomly divided intofour groups (eight mice per group), (I) exposure group: vehicle control group (control), treated with DMSO as negative control for 5 weeks; (II) exposure group: BPA exposure with dose of 50(L-BPA) or 500 μg/kg BW/day (H-BPA) for 5 weeks; (III) recovery group: vehicle control group treated with DMSO as negative control for 5 weeks + maintained with a standard chow diet for another 8 weeks to recover (R-control); recovery group: BPA exposure with doses of 50 or 500 μg/kg BW/day (BPA) for 5 weeks + maintained with a standard chow diet for another 8 weeks to recover (R-BPA). The above administration is by gavage. The maximum exposure dose for this topic meets the Food and Drug Administration (FDA) safety limits for BPA doses. 

### 2.3. Cell Culture and Treatment

Normal mouse hepatocytes (alpha mouse liver 12) AML12 were cultured in a high glucose DMEM basic medium containing 10% FBS (Gibco, Waltham, MA, USA) and a solution of 1% penicillin and streptomycin (PS) (Sigma, St. Louis, MO, USA). Cells were treated with DMSO and BPA at a concentration of 30 μM for two days, respectively. Plasmid transfection was then performed using Lip3000 liposome transfection reagent (Sigma, St. Louis, MO, USA). After two days of continued drug treatment, the following experimental manipulations were performed.

### 2.4. Measurements

Body weight and food intake of the animals were measured weekly. At the end of the study, mice were sacrificed by CO_2_ inhalation after overnight fasting. The body weights of the mice were counted after fasting, and the liver tissue and epididymal adipose tissue were collected and counted separately. Fat to body weight is the ratio of the absolute weight of epididymal fat to body weight. Blood was collected from the heart, and serum was obtained by centrifugation at 3000 rpm for 20 min for subsequent analysis. Serum total cholesterol (TCHO), triglyceride (TG), high-density lipoprotein cholesterol (HDL-C), low-density lipoprotein cholesterol (LDL-C), alanine amioTransferase (ALT) and aspartate Transaminase (AST) were measured by using biochemical assay kits provided by the Jiancheng Bioengineering Institute (Nanjing, China). All assays were performed according to the instructions supplied by the manufacturer. 

### 2.5. Histological Analysis

Freshly harvested adipose tissue of the epididymis and liver was collected, then washed with phosphate-buffered saline (PBS), and the tissues were immediately soaked in 4% of paraformaldehyde for 24 h. After dehydration, the specimens were intactly embedded in paraffin and sectioned (5 μm) for H&E staining. The morphological changes in all the slices in the liver and adipose tissue were examined with a high-resolution CS2 slide scanner (Leica Biosystems Inc., Buffalo Grove, IL, USA).

### 2.6. RT-qPCR Analysis

Total RNA Isolation Reagent was used to extract total RNA from liver tissues and reverse-transcribed into cDNA by a reverse transcription kit (TransGen, Beijing, China). The key genes were amplified using a SYBR-Green kit following the manufacturer’s instruction on a LightCycler480 (Roche, Shanghai, China). The amplification procedure was as follows: 95 °C for 8 min; 95 °C for 10 s, 60 °C for 20 s, 72 °C for 20 s (36 cycles), and final extension at 4 °C. All of the primers for the genes are shown in the Appendix A. The relative expression levels of genes were normalized using *18S* and the fold changes were analyzed using the 2^−ΔΔct^ method.

### 2.7. Western Blot Analyses

Total proteins of livers in different groups were harvested using RIPA buffer (Sigma-Aldrich, St. Louis, MO, USA) supplemented with a protease/phosphatase inhibitor (Sigma-Aldrich, St. Louis, MO, USA). Protein concentrations were then determined with the BCA protein detection kit (Thermo Fisher Scientific, Beijing, China). A 20 μg sample of protein was separated by 10% SDS-PAGE gels and then electrically transferred onto PVDF membranes (Millipore, Boston, MA, USA). The membranes were then blocked for 2 h with solutions containing 5% of non-fat milk. After washing, the membranes were incubated with a primary antibody (Abcam, Cambridge, MA, USA) overnight at 4 °C and then incubated with a labeled secondary antibody (Boster, Pleasanton, CA, USA) for 1 h at room temperature. Protein bands were visualized via Image J software, version 6.1 (Bio-Rad, Hercules, CA, USA). The *GAPDH* level was used as a loading control. 

### 2.8. Statistical Analysis

Data shown were expressed as mean ± standard deviation of all independent experiments (number of parallel ≥ 3) and one-way analysis of variance (ANOVA) with Duncan’s multiple range tests were used to compare the different treatment mice groups through SPSS 19.0 software (SPSS Inc., Chicago, IL, USA). A *p* value < 0.05 was considered to be statistically significant.

## 3. Results

### 3.1. BPA Exposure Induced Obesity in Mice

The mice from the control and BPA groups were sacrificed at the age of 8 weeks (BPA exposure) and at the age of 16 weeks (BPA exposure for 8 weeks + recovery for 8 weeks), respectively. During the exposure period, compared with the control group, the body weight (*p* < 0.05) and fat-to-body weight ratio (*p* < 0.01) of L-BPA and H-BPA groups were significantly increased. During recovery, we found that there was a significant decrease in the body weight and fat-to-body weight ratio of the male mice between the BPA group and the control group (*p* < 0.05) (Figure 1A,B). Furthermore, as shown in Figure 1C, adipocytes had a clearly larger size and volume both for BPA exposure and recovery groups than those in the vehicle control group. Interestingly, the size and volume of adipocytes were still larger even if at the recovery time (Figure 1D).

### 3.2. BPA Exposure Altered the Homeostasis of Metabolic Outcomes in Mice

After BPA treatment for 8 weeks, compared with the corresponding control groups, serum HDL, LDL, TCHO and TG levels were not obviously reduced in the 50 μg/kg BW/day and 500 μg/kg BW/day-exposed groups, whereas serum LDL and TG levels were only markedly decreased in the 500 μg/kg BW/day exposed group (Figure 2A–D). Importantly, as shown in Appendix A, compared with the corresponding control groups, serum APOD levels for 8 weeks were higher in the BPA exposure, whereas their levels decreased rapidly for another 8 weeks only to recover.

### 3.3. BPA Exposure Disrupted Hepatic Lipid Metabolism in Mice

#### 3.3.1. BPA Exposure Induced Liver Injury

The relative liver weights were markedly decreased in male mice after exposure to 500 μg/kg BW/day compared with the control (Figure 3A); during the recovery period, we also found that the high dose group resulted in a significant reduction in the liver weights of the mice (Figure 3B). The data of H&E-stained liver sections revealed normal liver histology in the control group manifesting as normal cell size, with uniform cytoplasm, and a prominent cell nucleus. Meanwhile, liver tissue sections failed to show evidence of lipid droplets or other aberrant changes, such as degeneration or necrosis (Figure 3C). Conversely, the mice exposed to BPA fed on the same diet showed liver pathologies seen in human NAFLDs, including small vacuoles, disordered hepatic cell cords, and increased fat deposition. For the recovery groups, an evident injury was observed in the livers of the mice in both the L-BPA and H-BPA groups. But notably, fat in the liver was mainly observed as macrovesicular droplets in the mice exposed to low-dose BPA (Figure 3D). The quantitative TG assay showed that BPA exposure resulted in a significant increase in TG levels in mouse livers in both the L-BPA and H-BPA groups (Figure 3E). However, this increase in attenuated during recovery (Figure 3F).

#### 3.3.2. BPA Exposure Altered the Genes Expression in Male Mice

Our previous study used RNA-Seq to explore potential mechanisms of BPA-induced adipogenesis in 3T3-L1 preadipocytes and a series of genes associated with de novo lipogenesis and lipid transport were detected to be regulated by BPA [11]. Based on fold change, here we examined the expression patterns of genes *APOD*, *SCD1*, *ANGPT4*, *LPL*, *G0S2*, *FADS2*, *GNAI2, PLIN1*, *ELOVL6*, *ACSL3*, *PPARα*, *PPARβ*, *PPARγ*, *FADS1* and *SOD3* in the livers of mice for BPA exposure and recovery groups. Importantly, we also found a significant expression reversion for stearoyl-CoA desaturase 1 (*SCD1*) (*p* < 0.01), apolipoprotein D (*APOD*) (*p* < 0.01), *ANGPT4* (*p* < 0.01), *PPARβ* (*p* < 0.05), *LPL* (*p* < 0.01) and *G0S2* (*p* < 0.01) between the exposure and recovery groups (*p* < 0.05), especially for *SCD1* and *APOD* (Figure 4A). Meanwhile, as shown in Appendix A, BPA can also decrease the expression level of *PPARα*, which was responsible for fatty acid oxidation. Besides, BPA can also increase mRNA expression levels of genes for lipogenesis, such as *FADS1*, *FADS2*, *LPL*, *G0S2*, and *ACSL3*.

#### 3.3.3. BPA Exposure Induced Hepatic Inflammation

In comparison to the control group, the expression levels of liver *TNF-α, IL-6* had an obvious increase in the animals exposed to BPA (*p* < 0.01) (Figure 4D). Whereas, the level of *IL-1β* were not increased significantly (*p* > 0.05). After 8 weeks of recovery, the degree of damage gradually decreased. As shown in Figure. 4D, during the recovery period, the expression levels of these genes returned to their initial levels except for *SAA3* (*p* < 0.05).

#### 3.3.4. Effects of BPA Exposure and Recovery on the Expression of SCD1 and APOD 

The expression of the proteins (*SCD1*, and *APOD*) were further determined by western blotting for the exposure and recovery groups. The results indicated that the levels of *SCD1* had a significant increase in BPA groups (*p* < 0.05). However, during recovery, there was no significant difference in the *SCD1* protein levels between the BPA group and control group (*p* > 0.05) (Figure 4B). Notably, BPA exposure groups had a lower level of APOD protein level (*p* = 0.097) and accordingly, there was an extremely significant increase after stopping exposure (Figure 4C).

### 3.4. Effect of APOD Over-Expression on BPA-Induced Dysregulation of Lipid Homeostasis

To assess the potential role of *APOD* in BPA-induced lipid disorders, we investigated whether BPA could increase fat accumulation by regulation of *APOD* in AML12 cells. As shown in Figure 5D,E, TG quantitative assay and oil red O staining results demonstrated that BPA exposure increased TG content, while *APOD* over-expression suppressed TG accumulation caused by BPA in the AML12 cells. These positive effects of *APOD* may be associated with upregulation of *PPARβ*, which plays important roles in angiogenesis, metabolism, and inflammation. These results for the first time demonstrate that *APOD* regulates BPA-induced dysregulation of lipid homeostasis. In summary, as shown in Figure 5, our findings verified that BPA aggravated lipid accumulation in hepatocytes via decreasing *APOD* expression in vitro and in vivo. *APOD* is a potential effective agent for the treatment of obesity and NAFLD caused by BPA exposure.

## 4. Discussion

Epidemiological and experimental studies suggested that the prevalence of NAFLDs may be associated with BPA exposure [12,13], but the mechanism is unclear. Our data demonstrated that the development of NAFLDs induced by BPA exposure was associated with hepatic pro-inflammatory, abnormal lipid metabolism and lipid deposition. Consistent with these results obtained by previous studies [14,15,16] our study showed that BPA induced hepatic steatosis and fat accumulation [11] revealing that BPA resulted in dose-dependent effects on metabolic parameters [17]. Our study shows that BPA exposure causes weight gain and liver lipoatrophy in mice. Therefore, exposure to BPA in early life should be carefully examined in the etiology of NAFLDs.

As is known, high or prolonged exposure to BPA during early life may exert more long-term adverse outcomes, and increase the risk factors associated with metabolic diseases in adult life [7]. During the exposure period, compared with the control group, the body weight and fat-to-body weight ratio of BPA groups were significantly increased. During the recovery process, we found that there were no significant differences between the BPA exposure group and the control group after the removal of BPA, indicating the recovery of the lipid metabolism. However, the size and volume of adipocytes were still larger even if at the recovery time and the possible cause was irreparable damage to the lipid metabolism. Meanwhile, significant changes in males in serum THCO and HDL-C levels were not found during BPA exposure and post-exposure recovery. But it is worth noting that TG and LDL-C were still higher in low-dose BPA groups, whether during BPA exposure or the recovery period. Meanwhile, it is an interesting finding that fat in the liver was mainly observed as macrovesicular droplets in the mice exposed to low-dose BPA. These data suggested that the live function recovered faster in the high-dose BPA rather than the low-dose BPA. The results in this study are consistent with previous studies that the mice are more sensitive to low-dose BPA exposure as compared to higher doses, thereby contributing to hepatic steatosis [9,14]. We cannot rule out the possibility that other mechanisms, whereas we speculated the dysregulated autophagy by BPA may contribute to the transcriptional impacts of low BPA doses reported here [18]. This raises further questions regarding whether the high-dose BPA-caused impairment to the lipid metabolism mechanism in male mice is identical to that of low-dose BPA.

Our RNA-seq analysis detected some key genes involved in TG and lipid metabolism, including *APOD*, *SCD1*, *ANGPT4*, *LPL*, *G0S2*, *FADS2*, *GNAI2*, *PLIN1*, *ELOVL6*, *ACSL3*, *PPARα*, *PPARβ*, *PPARγ*, *FADS1* and *SOD3* [11]. In this study, we further explored the changes in these genes during BPA exposure and post-exposure recovery. BPA can increase the mRNA expression levels of *FADS1*, *FADS2*, *LPL*, *G0S2*, and *ACSL3*, which were key regulators of lipogenesis [19,20]. Lipid accumulation in liver could be due to the different expression levels of these genes [21,22]. Interestingly, the mRNA expression levels of all these genes nearly reached the control levels during the recovery process, indicating that the recovery speeds of these genes were similar. Meanwhile, these genes were easier to recover in the liver and we believe that the recoveries of different genes follow a specific order after the removal of BPA exposure. This restoration mechanism ensures the recovery of liver function.

Importantly, we found a significant expression reversion for *SCD1*, *APOD*, *ANGPT4*, *PPARβ*, *LPL* and *G0S2* between the exposure and recovery groups, especially for *SCD1* and *APOD*. The mRNA and Western blotting results indicated that the levels of *SCD1* protein levels showed a remarkable increase in the liver of BPA groups while there was no significant difference between the BPA group and control group during recovery. Our results indicated that BPA can affect *SCD1* expression and further impair the process of murine adipogenesis [11]. The decreased expression of *SCD1* showed decreased hepatic triacylglycerol content and reduced body obesity, which was consistent with Zou et al. [23,24]. On the contrary, there was a significantly lower level of APOD protein in the exposure phase. However, in the recovery phase there was an extremely significant increase after exposure had ceased. Additionally, *APOD* over-expression suppressed TG accumulation in the AML12 cells. Therefore, *APOD* played an important role in BPA-induced dysregulation of lipid homeostasis and this increase may be an adaptive mechanism that ensures the recovery of liver function. 

Our results support the hypothesis that upregulation of APOD ameliorates dysregulation of lipid homeostasis caused by BPA exposure in male mice. Of relevance, our results showed that the expression of APODs was lower in the BPA group compared with control group mice and suggested an important role for APODs in regulating lipid metabolism caused by BPA exposure. BPA may decrease lipid translocation processes by inhibiting APOD expression in the liver. Interestingly, an important characteristic of BPA-induced hepatic steatosis was that compared to the control group, the trend of APOD concentration in the serum of mice in the BPA group was opposite to the trend of the level of APOD protein expression in the liver. This intriguing phenomenon suggests that the expression patterns are completely different in the liver and blood. Indeed, APOD appears to be a beneficial actor in both lipid metabolisms as it is associated with lipid uptake and inflammation resorption. However, how APOD controls its expression levels remains unknown and needs further attention.

## 5. Conclusions

In our study, we found that BPA caused severe steatosis in the livers of mice, which was partially alleviated after we stopped exposure. Importantly, BPA could significantly decrease the level of *APOD* protein whereas an extremely significant increase occurred after we stopped exposure. Meanwhile, *APOD* over-expression suppressed TG accumulation in AML12 cells. In conclusion, the damage caused by BPA can be repaired by upregulation of *APOD* and it is a potentially effective biochemical detection indicator for the treatment of obesity or NAFLDs caused by BPA exposure.

## Figures and Tables

**Figure 1 toxics-11-00775-f001:**
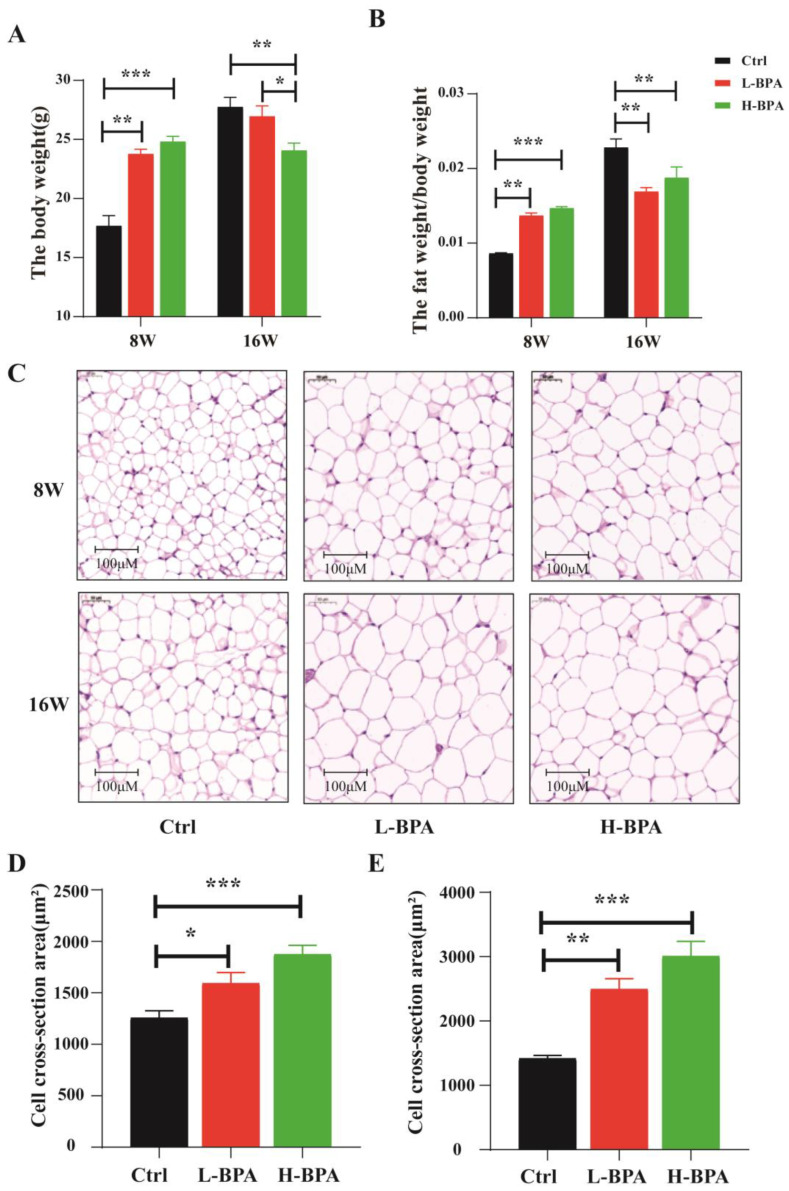
Effects of BPA exposure on the basic changes of parameters in male mice. Note: (**A**) Effects of BPA exposure on the body weight of mice; (**B**) Effects of BPA exposure on fat-to-body weight ratio; (**C**) Representative images of adipose tissue stained with H&E in mice for 8 weeks and 16 weeks (400×); (**D**) Relative area statistics of cells in the adipose tissue of the epididymis for 8 weeks. (**E**) Relative area statistics of cells in the adipose tissue of the epididymis for 16 weeks. Use ImageJ software (Bio-Rad, Hercules, CA, USA). *n* = 8 in each group. “8W” represents the control group treated with 2% of DMSO and the BPA group treated with BPA for 8 weeks, respectively. “16W” represents the control group treated with 2% of DMSO for 16 weeks and the BPA group treated with BPA for 8 weeks followed by an 8-week recovery period (cessation of drug exposure). Statistical significance was determined by one-way ANOVA. * Represents the significance at *p* < 0.05. ** Represents the significance at *p* < 0.01. *** Represents the significance at *p* < 0.001. L-BPA, 50 μg/kg BW/day BPA; H-BPA, 500 μg/kg BW/day BPA.

**Figure 2 toxics-11-00775-f002:**
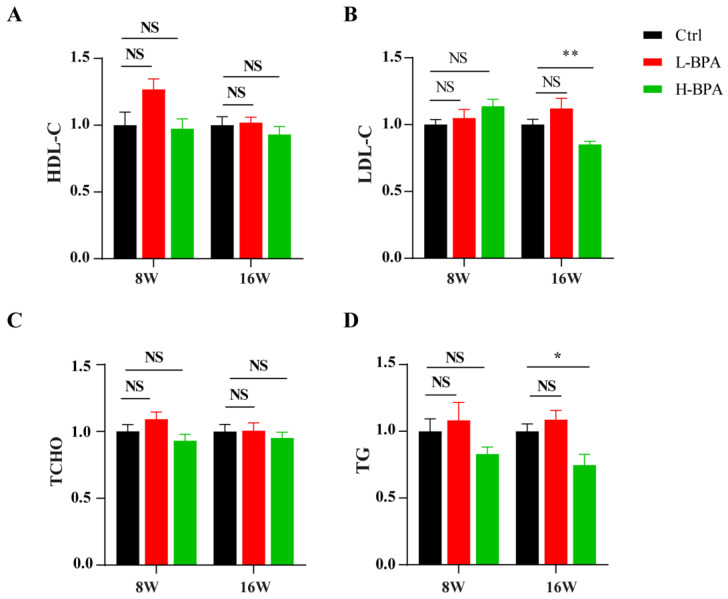
The blood biochemical indicators of mice. Note: (**A**) Relative serum high density lipoprotein cholesterol (HDL-C) levels; (**B**) Relative serum low density lipoprotein cholesterol (LDL-C) levels; (**C**) Relative serum total cholesterol (TCHO) levels; (**D**) Relative serum triglyceride (TG) levels. *n* = 6 in each group. Statistical significance was determined by one-way ANOVA. “8W” represents the control group treated with 2% DMSO and the BPA group treated with BPA for eight weeks, respectively. “16W” represents the control group treated with 2% DMSO for 16 weeks and the BPA group treated with BPA for eight weeks followed by an eight-week recovery period (cessation of drug exposure). * Represents the significance at *p* < 0.05. ** Represents the significance at *p* < 0.01. NS represents the significance at *p* > 0.05.

**Figure 3 toxics-11-00775-f003:**
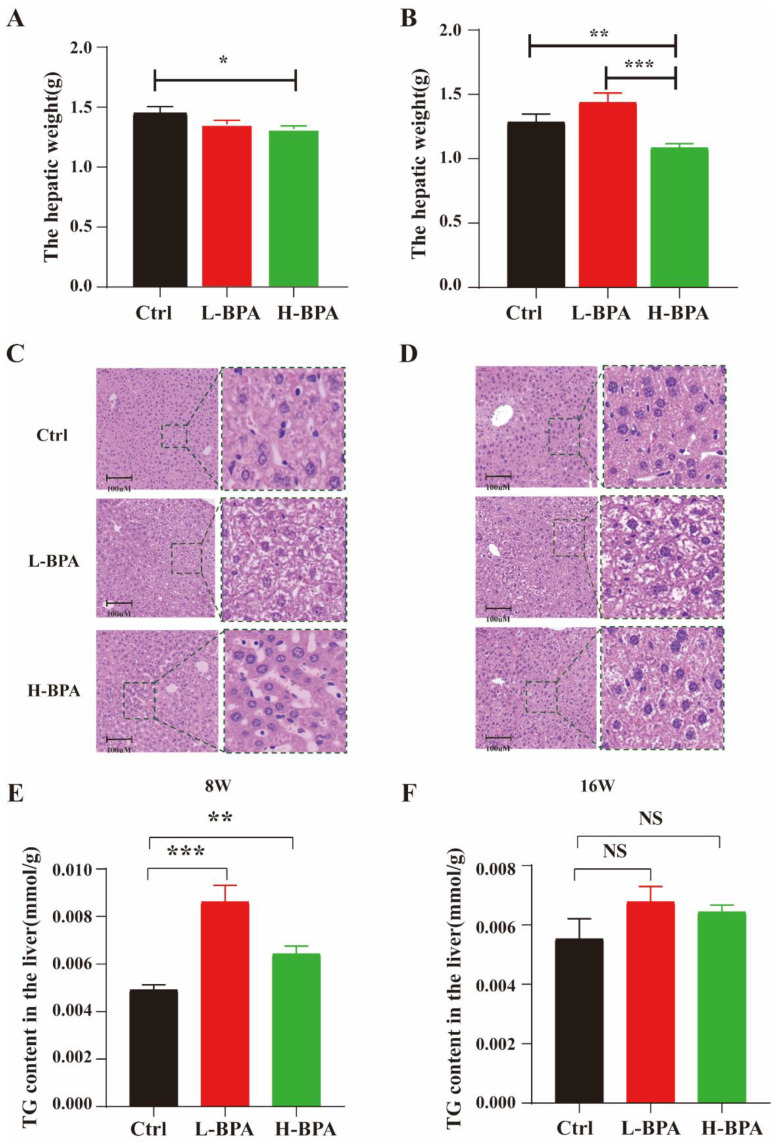
BPA-exposed mice exhibited hepatic lipid accumulation. Note: (**A**) Eight-week mouse liver weights; (**B**) Sixteen-week mouse liver weights; *n* = 8 in each group; (**C**,**D**) Representative images of liver with H&E in mice for 8 weeks and 16 weeks (400×). (**E**,**F**) TG content in mice liver in 8w and 16w groups, respectively. “8W” represents the control group treated with 2% DMSO and the BPA group treated with BPA for eight weeks, respectively. “16W” represents the control group treated with 2% DMSO for 16 weeks and the BPA group treated with BPA for eight weeks followed by an eight-week recovery period (cessation of drug exposure). Statistical significance was determined by one-way ANOVA. * Represents the significance at *p* < 0.05. ** Represents the significance at *p* < 0.01. *** Represents the significance at *p* < 0.001. NS represents the significance at *p* > 0.05.

**Figure 4 toxics-11-00775-f004:**
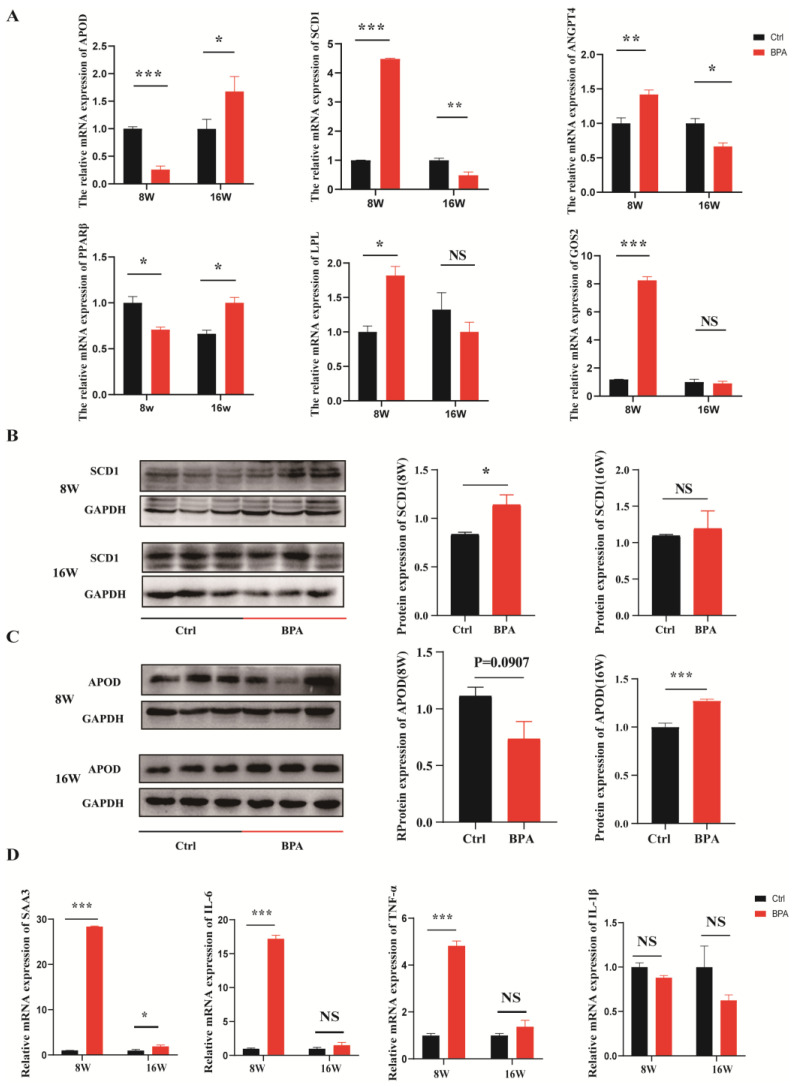
Effects of BPA Exposure on the of Gene Expressions in Liver. Note: (**A**) Effects of BPA on the mRNA expression of genes regulating lipid metabolism. *n* = 5 in each group; (**B**) Expression of SCD1 protein in mouse liver; (**C**) Expression of APOD protein in mouse liver; Quantitative analysis of protein expression was performed using the ImageJ software (Bio-Rad, Hercules, CA, USA); (**D**) Effects of BPA on the mRNA expression of genes regulating inflammatory, *n* = 5 in each group. “8W” represents the control group treated with 2% DMSO and the BPA group treated with BPA for eight weeks, respectively. “16W” represents the control group treated with 2% DMSO for 16 weeks and the BPA group treated with BPA for eight weeks followed by an eight-week recovery period (cessation of drug exposure). * Represents the significance at *p* < 0.05. ** Represents the significance at *p* < 0.01. *** Represents the significance at *p* < 0.001. NS represents the significance at *p* > 0.05.

**Figure 5 toxics-11-00775-f005:**
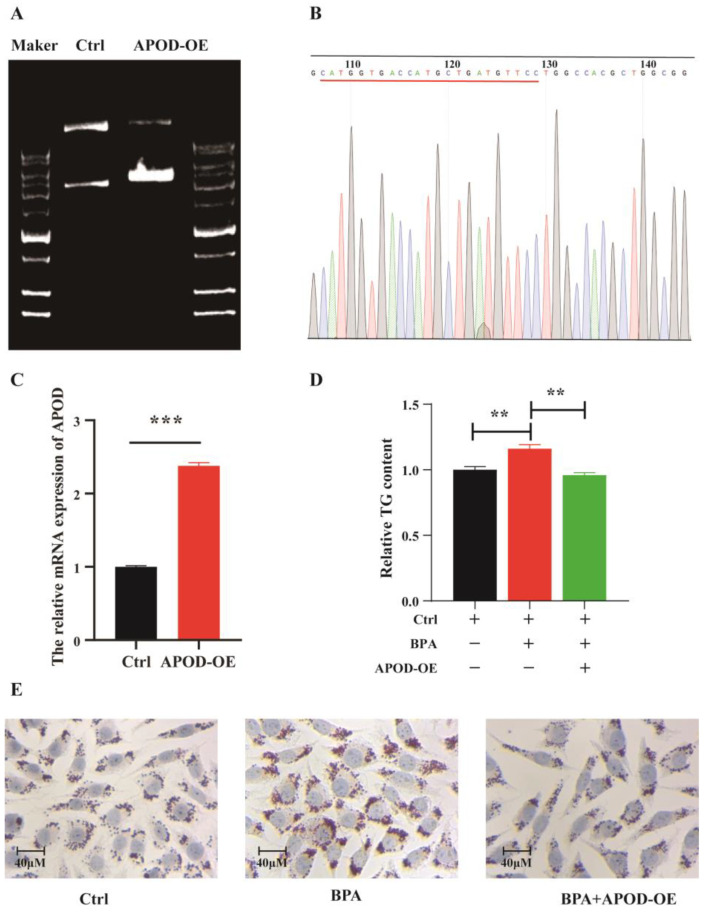
Effect of *APOD* over-expression on the TG accumulation in AML12 cells. Note: (**A**) Electrophoresis of pcDNA3.1 (+) APOD plastimid; (**B**) The sequencing of recombinant plasmid DNAs BPA exposure promoted hepatic inflammatory response in mice; (**C**) The expression of APOD in AML12 cells, *n* = 3 in each group; (**D**) Relative TG content in cells, *n* = 4 in each group; (**E**) Typical image of lipid accumulation in AML12 cells by oil red O staining. ** Represents the significance at *p* < 0.01. *** Represents the significance at *p* < 0.001.

## Data Availability

The data are available if requested.

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
