# Peer review of "Effects of BPA Exposure and Recovery on the Expression of Genes Involved in the Hepatic Lipid Metabolism in Male Mice"

_toxics, 2023, doi:10.3390/toxics11090775_

Round 1

Reviewer 1 Report

The manuscript from Li et al. aims to determine whether the negative effect of BPA exposure on liver lipid accumulation is reversible after a period with no BPA exposure. This is a very important question in this area of research. However, the poor quality of the text and data presented, as well as a poor study design takes away the excitement of the reviewer.

Major concerns:

1.     The design of the study could be improved: to conclude that the recovery period had an effect, it would have been important to have an additional group of mice treated for the 16 weeks with BPA. Indeed, it is possible for example that BPA exposure induces an increase in some markers during the first weeks of exposure because the body is trying to compensate, but prolonging the exposure could result in a decrease in the same markers (as seen for example for the mRNA expression of some markers). It is for example what would happen in the development of diabetes where insulin levels increase in pre-diabetes and then decrease over time. Thus, the fact that there are differences after the recovery period is difficult to interpret without having mice exposed to BPA for 16 weeks.

It is also difficult to conclude that the gene expression is reversed with the recovery period since there is no comparison between the control group at 8W and the control group at 16W. Indeed, it is possible that aging alters the expression of the genes independently from the exposure to BPA.

2.     Please provide more details in all your figure legends (explain how the data were collected, provide the n, explain the meaning of the stats signs, etc) The figure legend should help the reader to understand the figure without having to go to the method or result sections.

3.     Figure 1A: I am unsure to understand what is shown here. The figure legend seems to mention this is the change in body weight, but the y-axis legend only mentions: "The body weight". Also, there is no unit on the y-axis (true for all figures please add the units on the y-axes). If the change in body weight is shown, what is it exactly? Is this the change from the beginning of the study for all groups? It doesn't seem to be the case since the change is similar for the 8W and 16W control groups, and we probably expect to have a bigger weight gain after 16W than 8 W. It also seems that the control groups in figures 1A and 1B have a value of 1. Are the data presented relative to the control group? This seems an unusual way to present body weight or body weight gain. Also, the fat weight/body weight ratio can't have a value of 1 nor a value above 1 since it would mean that the mice had more fat weight than body weight (unless the data are presented relative to the control group, which again seems an unusual way to present this type of data). Please present the absolute values of body weight (or change in body weight from the beginning of the study) and the absolute values of fat weight/body weight.

4.     L137-138: By looking at the pictures, this is not clear that BPA increased the size of the adipocytes. On the contrary, it seems that adipocytes have the same size in all conditions but the one for H-BPA-W16, were the adipocytes seem smaller. There is a need to quantify the size of the adipocytes using an appropriate number of pictures per mouse in order to be able to draw conclusions here.

5.     Results presented in Figure 3.2 are not clear. In the method section and figure 2, it seems that HDL-, LDL-chol, total cholesterol and TG have been measured in blood samples, but in the result section , it seems to be in the liver. Please correct this inconsistency.

6.     L163-165 and Figure 3A-B: it is concluded that the decrease in liver weight demonstrates that BPA induces "hepatomegaly". Isn't hepatomegaly an increased in liver weight/size? Furthermore, since the data are presented as liver weight / body weight, but the mice on BPA showed an increased body weight (Figrue 1), it seems that the weight of the liver is maybe not that different in the control mice vs the BPA-treated mice. It would be more important to show the absolute liver weight than the liver/body weight ratio. Again, the data presented in figure 2 are strange since liver weight/body weight could not be of one or above one.

7.       Figure 3C-D: an increased lipid droplet accumulation with BPA is not obvious on some pictures. Would it be possible to measure liver TG accumulation using a more quantitative method?

8.     SCD1 Western blots for 8W and 16W mice seem to be very different: why does the W8 figure show 2 (almost 3) bands, but there are only one band in the 16W figure?

9.     The AML12 cell culture and treatments are not described in the method section. We thus don't know for how long the cells were treated with BPA and at which concentrations. Without this information, it is difficult to draw any conclusion from the results since it is not clear if the concentrations chosen are environmentally relevant.

10.  L216-217: “To assess the potential role of APOD on BPA-induced lipid disorders, we investigated whether BPA could increase fat accumulation by regulaton (typo here) of APOD in AML12 217 cells.” Since there is no description of the AML12 treatment, it is hard to understand what the exact experiment was, but my understanding is that cells were treated with BPA with or without APOD overexpression. This experiment doesn’t help to determine if BPA induces lipid accumulation in hepatocytes through APOD. Indeed, this experiment shows the effect of overexpressing APOD on lipid accumulation in BPA-treated adipocytes. To determine if APOD is involved in BPA-induced lipid accumulation, the opposite model should have been chosen: knockdown or knockout of APOD in cells treated with BPA.

11.   Figure 5 data are very difficult to interpret since TG in AML12 cells were only measured with a qualitative method and only one picture of each treatment is shown in the figure. It is important to quantitatively measure TG content with at least 3 independent experiments to be able to conclude that APOD overexpression decreases lipid content.

12.   L223-225: the conclusion drawn from figure 5 data are in contradiction with some results shown in the present manuscript: “our findings verified showed that BPA aggravated lipid accumulation in adipocyte and hepatocytes via increasing APOD expression in vitro and in vivo.” Firstly, figure 4 shows that BPA decreased (but not increased) APOD expression. Secondly, figure 5 shows that increasing APOD expression decreased (but not increased) lipid accumulation in hepatocytes. Lastly, APOD levels has not been measured in adipose tissue in the present study, thus, it is inappropriate to conclude that APOD is responsible for the increased lipid accumulation in adipocytes of mice treated with BPA (also, as mentioned earlier, it is not obvious adipocytes were bigger with BPA exposure).

13.   Discussion:

-          L239-240: It is mentioned that only the high BPA dose induced an increased steatosis and fat accumulation, but this conclusion is not obvious in figure 3.

-          L242-243: It is mentioned that BPA did not affect body weight, however, this is not what is shown in figure 1 and the opposite is discussed in the next paragraph.

-          L251-253: As mentioned earlier, it is not obvious that the adipocytes are bigger with BPA exposure with the provided data. This conclusion should be modified.

-          L293-297: The paper cited here (Do Carmo et al., 2008) doesn't look at the effect of APOD overexpression on liver steatosis / hepatic insulin resistance but at its neuroprotective effect with coronavirus OC43 infection. Furthermore, it is mentioned in the present discussion that mice overexpressing APOD develop steatosis and IR, but this is in opposition with the interpretation of the results of the current paper where it is discussed that overexpressing APOD decreased lipid content in hepatocytes.

-          L301-302: Which data suggest that APOD-induced lipid accumulation is due to increased lipid uptake? Also, it is in contradiction with the authors' conclusion that overexpression of APOD decreased lipid accumulation.

-          L302-304: I agree that APOD in the serum show the reverse of what was found in the liver, but the manuscript doesn’t show that those levels were inversely correlated. It would be important to run some statistics to be able to conclude this.

Minor concerns:

1.     Introduction: Recently, a new nomenclature to classify fatty liver diseases has been accepted (MASLD/MASH). DOI: 10.1097/HEP.0000000000000520. Please use this new nomenclature in your manuscript.

2.     L35-37: What do you mean here? My understanding is that we may expose children to environmental pollutants to better understand how they affect biological mechanisms. Isn't it unethical? It is also unclear why there is a focus on exposure to pollutants during early life in the introduction since this is not the focus of the present manuscript.

3.     L54: “TG” This abbreviation is not defined earlier. Please define all your abbreviations at the first use (and please, do not define them several times throughout the manuscript).

4.     Methods: Please provide more information about BPA exposure: were the mice exposed to BPA in their food/water or by injection? Please also justify the doses of BPA used: is it environmentally relevant or higher than what humans are exposed to daily.

5.     L95: "adipose tissues" please precise which type of adipose tissues was collected.

6.     L96: "fixed" is repeated twice in the sentence.

7.     L99: it is not clear why "colon morphology" is mentioned here since at the beginning of this section it was mentioned adipose tissues and liver were collected (not the colon) and there are also no results on colon morphology in the rest of the study.

8.     L113: a space is missing between with and the ("with the BCA protein") and then on L114, there is an extra space between "was" and "separated"

9.     L120: Image J is not a software from Bio-Rad but from the NIH.

10.   L133: This is the first time in the manuscript that L-BPA and H-BPA are mentioned. The method section does not describe these 2 doses of BPA. Please correct this oversight.

11.   L135, it is mentioned "male mice" However, in the method section, L74, it is mentioned that the female mice were exposed to BPA (not the male mice). This is confusing.

12.   L132-134: please reword the sentence (there is probably no need to say: "in the BPA group, respectively" at the end of the sentence)

13.   Figure 4: the figure legend and the citation in the text don't fit the figures (inflammation is shown in D in the figure but the legend and result section says B - same for protein expression data)

14.   L275: "These genes recovered faster in the liver" What do you mean? Faster compared to what?

The paper should be reviewed by an English-speaking person to make it easier to read.

Author Response

Dear editor:

Thank you very much for providing an opportunity for us to revise our paper. We also wish to thank you and reviewers for the positive comments and constructive suggestions on the manuscript, which greatly improve the manuscript. Accordingly, we have made the essential revisions suggested by the reviewers as follows.

Major concerns:

  1. The design of the study could be improved: to conclude that the recovery period had an effect, it would have been important to have an additional group of mice treated for the 16 weeks with BPA. Indeed, it is possible for example that BPA exposure induces an increase in some markers during the first weeks of exposure because the body is trying to compensate, but prolonging the exposure could result in a decrease in the same markers (as seen for example for the mRNA expression of some markers). It is for example what would happen in the development of diabetes where insulin levels increase in pre-diabetes and then decrease over time. Thus, the fact that there are differences after the recovery period is difficult to interpret without having mice exposed to BPA for 16 weeks.

Response: Thank you very much for your comments and I fully agree with you. Yes, it would have been important to have an additional group of mice treated for the 16 weeks with BPA. Our other study (unpublished) could provide evidence to support this opinion. We used a 16-week long-term BPA mouse exposure model (PND0-PND112) and found that the trends of hepatic steatosis and lipid metabolism-related genes in the liver under sustained BPA exposure were consistent with those under the 8-week drug exposure condition in this study.

It is also difficult to conclude that the gene expression is reversed with the recovery period since there is no comparison between the control group at 8W and the control group at 16W. Indeed, it is possible that aging alters the expression of the genes independently from the exposure to BPA.

Response: Thank you very much for your comments and I fully agree with you. The comparison of the 8W and 16W in the control group can show the effect of the age factor, so we used the comparison group (Control vs BPA (8W), R- Control vs R-BPA (16W)) to avoid the effects of age factor. At the same time, BPA was guaranteed to be the only variable in the manuscript.

  1. Please provide more details in all your figure legends (explain how the data were collected, provide the n, explain the meaning of the stats signs, etc) The figure legend should help the reader to understand the figure without having to go to the method or result sections.

Response: Thank you very much for your comments. As suggested, we have added more details in the figure notes in the manuscript.

  1. Figure 1A: I am unsure to understand what is shown here. The figure legend seems to mention this is the change in body weight, but the y-axis legend only mentions: "The body weight". Also, there is no unit on the y-axis (true for all figures please add the units on the y-axes). If the change in body weight is shown, what is it exactly? Is this the change from the beginning of the study for all groups? It doesn't seem to be the case since the change is similar for the 8W and 16W control groups, and we probably expect to have a bigger weight gain after 16W than 8 W. It also seems that the control groups in figures 1A and 1B have a value of 1. Are the data presented relative to the control group? This seems an unusual way to present body weight or body weight gain. Also, the fat weight/body weight ratio can't have a value of 1 nor a value above 1 since it would mean that the mice had more fat weight than body weight (unless the data are presented relative to the control group, which again seems an unusual way to present this type of data). Please present the absolute values of body weight (or change in body weight from the beginning of the study) and the absolute values of fat weight/body weight.

Response: Thank you very much for your comments. As suggested, we have revised the Y-axis legend of Fig1 A to "The body weight(g)" and the absolute values of the Fig1 A and Fig1 B in the manuscript.

  1. L137-138: By looking at the pictures, this is not clear that BPA increased the size of the adipocytes. On the contrary, it seems that adipocytes have the same size in all conditions but the one for H-BPA-W16, were the adipocytes seem smaller. There is a need to quantify the size of the adipocytes using an appropriate number of pictures per mouse in order to be able to draw conclusions here.

Response: Thank you very much for your comments. As suggested, we have revised the typical adipose tissue pathology section images and quantitatively measured the area of adipose tissue cells in the Fig1.

  1. Results presented in Figure 3.2 are not clear. In the method section and figure 2, it seems that HDL-, LDL-chol, total cholesterol and TG have been measured in blood samples, but in the result section, it seems to be in the liver. Please correct this inconsistency.

Response: Thank you very much for your comments. As suggested, we have rephrased Section 3.2 and added more details.

  1. L163-165 and Figure 3A-B: it is concluded that the decrease in liver weight demonstrates that BPA induces "hepatomegaly". Isn't hepatomegaly an increased in liver weight/size? Furthermore, since the data are presented as liver weight / body weight, but the mice on BPA showed an increased body weight (Figrue 1), it seems that the weight of the liver is maybe not that different in the control mice vs the BPA-treated mice. It would be more important to show the absolute liver weight than the liver/body weight ratio. Again, the data presented in figure 2 are strange since liver weight/body weight could not be of one or above one.

Response: Thank you very much for your comments. As suggested, we have revised the expression form of Figure 3A and 3B to use absolute weight to indicate the change in liver weight under different treatment conditions. In addition, we have removed the misleading phrase "suggests BPA exposure induces hepatomegaly".

  1. Figure 3C-D: an increased lipid droplet accumulation with BPA is not obvious on some pictures. Would it be possible to measure liver TG accumulation using a more quantitative method?

Response: Thank you very much for your comments. As suggested, we have added a quantitative TG assay of the liver for a better representation of the lipid content of the liver.

  1. SCD1 Western blots for 8W and 16W mice seem to be very different: why does the W8 figure show 2 (almost 3) bands, but there are only one band in the 16W figure?

Response: Thank you very much for your comments. The results of WB are influenced by too many factors. Here, the major cause maybe the individual differences in students. To be honest, our previous study (8 weeks) was finished by Ruyue Fang and the current study (16 weeks) and integrated analysis was finished by Changqing Li, so the results were very different. Based on these results, we have tried our best to resolve the problem and we were repeating this experiment and we hope we can get the perfect results as soon as possible.

  1. The AML12 cell culture and treatments are not described in the method section. We thus don't know for how long the cells were treated with BPA and at which concentrations. Without this information, it is difficult to draw any conclusion from the results since it is not clear if the concentrations chosen are environmentally relevant.

Response: Thank you very much for your comments. As suggested, we have added more details for the treatment of the cells as follows: AML12 were cultured in high glucose DMEM basic medium containing 10% FBS (Gibco, Waltham, MA, USA) and a solution of 1% penicillin and streptomycin (PS) (Sigma, St. Louis, MO, USA). Cells were treated with DMSO and BPA at a concentration of 30 μM for two days, respectively. Plasmid transfection was then performed using Lip3000 liposome transfection reagent (Sigma, USA). After two days of continued drug treatment, the following experimental manipulations were performed.

  1. L216-217: “To assess the potential role of APOD on BPA-induced lipid disorders, we investigated whether BPA could increase fat accumulation by regulaton (typo here) of APOD in AML12 217 cells.” Since there is no description of the AML12 treatment, it is hard to understand what the exact experiment was, but my understanding is that cells were treated with BPA with or without APOD overexpression. This experiment doesn’t help to determine if BPA induces lipid accumulation in hepatocytes through APOD. Indeed, this experiment shows the effect of overexpressing APOD on lipid accumulation in BPA-treated adipocytes. To determine if APOD is involved in BPA-induced lipid accumulation, the opposite model should have been chosen: knockdown or knockout of APOD in cells treated with BPA.

Response: Thank you very much for your comments. As suggested, we have added the detail for the cell treatments in the manuscript. Meanwhile, we found that BPA exposure resulted in hepatic lipid accumulation and reduced APOD expression in the liver of mice, and therefore the potential mechanism of APOD in the process of hepatic lipid degeneration due to BPA exposure was explored by reducing APOD expression in hepatocytes.

  1. Figure 5 data are very difficult to interpret since TG in AML12 cells were only measured with a qualitative method and only one picture of each treatment is shown in the figure. It is important to quantitatively measure TG content with at least 3 independent experiments to be able to conclude that APOD overexpression decreases lipid content.

Response: Thank you very much for your comments. As suggested, we have added the quantitative TG assays to illustrate changes in intracellular lipid content in the manuscript.

  1. L223-225: the conclusion drawn from figure 5 data are in contradiction with some results shown in the present manuscript: “our findings verified showed that BPA aggravated lipid accumulation in adipocyte and hepatocytes via increasing APOD expression in vitro and in vivo.” Firstly, figure 4 shows that BPA decreased (but not increased) APOD expression. Secondly, figure 5 shows that increasing APOD expression decreased (but not increased) lipid accumulation in hepatocytes. Lastly, APOD levels has not been measured in adipose tissue in the present study, thus, it is inappropriate to conclude that APOD is responsible for the increased lipid accumulation in adipocytes of mice treated with BPA (also, as mentioned earlier, it is not obvious adipocytes were bigger with BPA exposure).

Response: Thank you very much for your comments. As suggested, we have corrected two descriptive errors in the part 3.2 for the specific changes. Additionally, we have some changes to compensate the poor quality of the images in Figure 5E.

  1. Discussion:

-          L239-240: It is mentioned that only the high BPA dose induced an increased steatosis and fat accumulation, but this conclusion is not obvious in figure 3.

Response: Thank you very much for your comments. As suggested, we have rephrased the sentences and added more details in the discussion section.

-          L242-243: It is mentioned that BPA did not affect body weight, however, this is not what is shown in figure 1 and the opposite is discussed in the next paragraph.

Response: Thank you very much for your comments. As suggested, we have corrected the incorrect statement to " Our study shows that BPA exposure causes weight gain and liver lipoatrophy in mice" and added more details in the discussion section.

-          L251-253: As mentioned earlier, it is not obvious that the adipocytes are bigger with BPA exposure with the provided data. This conclusion should be modified.

Response: Thank you very much for your comments. As suggested, we have revised Figure 1, which is consistent with the existing conclusions.

-          L293-297: The paper cited here (Do Carmo et al., 2008) doesn't look at the effect of APOD overexpression on liver steatosis / hepatic insulin resistance but at its neuroprotective effect with coronavirus OC43 infection. Furthermore, it is mentioned in the present discussion that mice overexpressing APOD develop steatosis and IR, but this is in opposition with the interpretation of the results of the current paper where it is discussed that overexpressing APOD decreased lipid content in hepatocytes.

Response: Thank you very much for your comments. As suggested, we have replaced the right reference in the manuscript.

-          L301-302: Which data suggest that APOD-induced lipid accumulation is due to increased lipid uptake? Also, it is in contradiction with the authors' conclusion that overexpression of APOD decreased lipid accumulation.

Response: Thank you very much for your comments. The suggestion that "APOD-induced lipid accumulation is due to increased lipid uptake" is a speculation based on the fact that APOD itself functions as an atypical apolipoprotein. However, our wording is not rigorous, so we changed it to " BPA may decrease lipid translocation processes by inhibiting APOD expression in the liver ". Details of the changes are given in the discussion for the specific changes.

-          L302-304: I agree that APOD in the serum show the reverse of what was found in the liver, but the manuscript doesn’t show that those levels were inversely correlated. It would be important to run some statistics to be able to conclude this.

Response: Thank you very much for your comments. The suggestion that " those levels were inversely correlated " is a speculation based on the fact that APOD itself functions as an atypical apolipoprotein. However, our wording is not rigorous, so we changed it to “Interestingly, an important characteristic of BPA-induced hepatic steatosis was that compared to the control group, the trend of APOD concentration in the serum of mice in the BPA group was opposite to the trend of the level of APOD protein expression in the liver.”.

Minor concerns:

  1. Introduction: Recently, a new nomenclature to classify fatty liver diseases has been accepted (MASLD/MASH). DOI: 10.1097/HEP.0000000000000520. Please use this new nomenclature in your manuscript.

Response: Thank you very much for your comments. In relation to the diagnostic criteria for MAFLD, it was found that the animal model in this subject may not fully meet the diagnostic criteria for MAFLD, therefore the animal disease phenotype was chosen to be defined as NAFLD.

  1. L35-37: What do you mean here? My understanding is that we may expose children to environmental pollutants to better understand how they affect biological mechanisms. Isn't it unethical? It is also unclear why there is a focus on exposure to pollutants during early life in the introduction since this is not the focus of the present manuscript.

Response: Thank you very much for your comments. On the basis of the reviewer's suggestion, we also consider this misrepresentation to be meaningless and have therefore removed “Children are vulnerable to environmental exposures, so these exposures can be used as tools to better study the biological mechanisms by which hepatic lipid accumulation occurs.”.

  1. L54: “TG” This abbreviation is not defined earlier. Please define all your abbreviations at the first use (and please, do not define them several times throughout the manuscript).

Response: Thank you very much for your comments. As suggested, we have defined all your abbreviations in the manuscript.

  1. Methods: Please provide more information about BPA exposure: were the mice exposed to BPA in their food/water or by injection? Please also justify the doses of BPA used: is it environmentally relevant or higher than what humans are exposed to daily.

Response: Thank you very much for your comments. As suggested, we have added more details in the manuscript.

  1. L95: "adipose tissues" please precise which type of adipose tissues was collected.

Response: Thank you very much for your comments. As suggested, we have added more details in the manuscript.

  1. L96: "fixed" is repeated twice in the sentence.

Response: Thank you very much for your comments. As suggested, we have rephrased the sentences.

  1. L99: it is not clear why "colon morphology" is mentioned here since at the beginning of this section it was mentioned adipose tissues and liver were collected (not the colon) and there are also no results on colon morphology in the rest of the study.

Response: Thank you very much for your comments. As suggested, we have added more details for the “Histological Analysis” in the manuscript.

  1. L113: a space is missing between with and the ("with the BCA protein") and then on L114, there is an extra space between "was" and "separated"

Response: Thank you very much for your comments. As suggested, we have rephrased the sentences.

  1. L120: Image J is not a software from Bio-Rad but from the NIH.

Response: Thank you very much for your comments. As suggested, we have rephrased the sentences.

  1. L133: This is the first time in the manuscript that L-BPA and H-BPA are mentioned. The method section does not describe these 2 doses of BPA. Please correct this oversight.

Response: Thank you very much for your comments. As suggested, we have added more details in the manuscript.

  1. L135, it is mentioned "male mice" However, in the method section, L74, it is mentioned that the female mice were exposed to BPA (not the male mice). This is confusing.

Response: Thank you very much for your comments. As suggested, we have rephrased the sentences.

  1. L132-134: please reword the sentence (there is probably no need to say: "in the BPA group, respectively" at the end of the sentence)

Response: Thank you very much for your comments. As suggested, we have rephrased the sentences.

  1. Figure 4: the figure legend and the citation in the text don't fit the figures (inflammation is shown in D in the figure but the legend and result section says B - same for protein expression data)

Response: Thank you very much for your comments. As suggested, we have revised the figure legend and the citation in the text.

  1. L275: "These genes recovered faster in the liver" What do you mean? Faster compared to what?

Response: Thank you very much for your comments. In our opinion, the remission of hepatic steatosis in the H-BPA group after cessation of BPA exposure was more pronounced than the recovery in the L-BPA group. The results in this study were consistent with previous studies, which showed that the mice were more sensitive to low-dose BPA exposure as compared to higher doses, thereby contributing to hepatic steatosis.

Thank you again for your comments. If you have any questions, please let me know as soon as possible.

Sincerely

Shaohua Yang or Hui-Li Wang

College of Food and Biological Engineering

Hefei University of Technology

193 Tunxi Road, Hefei 230009, Anhui, China

Tel/Fax: +86-551-65785485

Email: [email protected] or [email protected].

Reviewer 2 Report

Dear authors!

The aim of the research is very good, but processing of the data is inadequate. If your aim is to assess recovery you should make statistical processing in another way. First of all you mast evaluate differences between 16 weeks and 8 weeks in all the investigated parameters. 

In Materials and methods:

You need to justify the choice of doses of Bisphenol A.

Please, add information on methods of detection of cholesterol, triglyceride etc.

Calculation of fat-to-body weight ratio mentioned in the Results, is not described in Matterials and methods.

The assertion of an increase in the size of adipocytes is not supported by any data. Please note that photomicrographs of 3-4 cells are not evidence of an increase in size. Histomorphometric studies should be carried out.

You demonstrated decreased relative liver weight in Fig.3A. Please, explain why reduction of relative liver weihgt you considered a sign of hepatomegaly (lines 165-166)? Decreased relative liver weight is clearly resulted from increased body weight. In this case absolute liver weight  should be demonstrated.

I found no conclusion in yor paper.

Author Response

The aim of the research is very good, but processing of the data is inadequate. If your aim is to assess recovery you should make statistical processing in another way. First of all you mast evaluate differences between 16 weeks and 8 weeks in all the investigated parameters. 

Response: Thank you very much for your comments and I fully agree with you. Yes, we counted phenotypic metrics such as body weight, fat/body weight ratio, and liver mass by means of absolute values in order that it could be shown that the phenotypic differences were in the eight-week and sixteen-week groups of mice. Details of the changes are given in the Fig.1 and Fig. 3.

In Materials and methods:

You need to justify the choice of doses of Bisphenol A.

Response: Thank you very much for your comments. A maximum BPA dose of 500μg/kg/day was chosen, which is in compliance with the Food and Drug Administration (FDA) 2014 safety limits for BPA. The 2014 FDA safety limit for BPA is set at 500μg/kg/day, and in addition, the dose conversion factor between humans and mice is about 12.3, so we chose a dose of 500μg/kg/day of the drug.

Please, add information on methods of detection of cholesterol, triglyceride etc.

Response: Thank you very much for your comments. Lipid-related markers are measured by commercial kits (Nanjing, China). The basic principle is as follows: e.g. TG assay, where the lipid component combines with the detection reagent to form a coloured substance, quinone, and the lipid content is determined by measuring the absorbance of the sample.

Calculation of fat-to-body weight ratio mentioned in the Results, is not described in Matterials and methods.

Response: Thank you very much for your comments. As suggested, we have added some content to the “measurements” section based on the reviewers' comments, as follows: The body weights of the mice were counted after fasting, and liver tissue and epididymal adipose tissue were collected and counted separately. Fat to body weight is the ratio of the absolute weight of epididymal fat to body weight.

The assertion of an increase in the size of adipocytes is not supported by any data. Please note that photomicrographs of 3-4 cells are not evidence of an increase in size. Histomorphometric studies should be carried out.

Response: Thank you very much for your comments. Yes, we used the ImageJ software (National Institutes of Health, USA) to quantify the cross-sectional area of cells in the adipose tissue of the mouse epididymides, n=4 in each group. Details of the changes are shown in Fig 1.

You demonstrated decreased relative liver weight in Fig.3A. Please, explain why reduction of relative liver weihgt you considered a sign of hepatomegaly (lines 165-166)? Decreased relative liver weight is clearly resulted from increased body weight. In this case absolute liver weight  should be demonstrated.

Response: Thank you very much for your comments and I fully agree with you.  Mouse liver weights are expressed in absolute quantitative terms and detailed modifications are shown in Fig 3A.

I found no conclusion in yor paper.

Response: Thank you very much for your comments. As suggested, we have added conclusion in the manuscript.

Reviewer 3 Report

Review of the manuscript entitled: Effects of BPA exposure and recovery on the expression of genes involved in the hepatic lipid metabolism in male mice. The manuscript deals with a very important topic, threats such as BPA or EDCs are extremely dangerous.

1.      In abstract and introduction clear aim of the manuscript should be added e.g. "The aim of the present study was to ...". In case of introduction aim should be at the end of introduction.

2.      The reference style should be corrected, it is currently incorrect in the text. Moreover, in line 81 the year of the citation is missing.

3.      It should be noted that the methodology should include catalog numbers of antibodies (and their dilutions), catalog numbers of primers (or their sequences if they were designed).

4.      Was the study approved by the ethics committee? The acceptance number should be provided.

5.      The description of the results is correct, but it could be written how much exactly the change in the expression of genes and proteins is.

6.      The discussion is satisfactory. Lines 308-309 should be deleted or  authors should add conclusions.

Author Response

Dear editor:

Thank you very much for providing an opportunity for us to revise our paper. We also wish to thank you and reviewers for the positive comments and constructive suggestions on the manuscript, which greatly improve the manuscript. Accordingly, we have made the essential revisions suggested by the reviewers as follows.

Review of the manuscript entitled: Effects of BPA exposure and recovery on the expression of genes involved in the hepatic lipid metabolism in male mice. The manuscript deals with a very important topic, threats such as BPA or EDCs are extremely dangerous.

  1. In abstract and introduction clear aim of the manuscript should be added e.g. "The aim of the present study was to ...". In case of introduction aim should be at the end of introduction.

Response: Thank you very much for your comments and I fully agree with you. As suggested, we have added more details in the abstract and introduction sections. In the “Abstract” we added: Therefore, this project aims to investigate the effects of BPA on hepatic lipid metabolism function and its potential mechanisms in mice by comparing the BPA exposure model and the BPA exposure and cessation of drug treatment model. In the “Introduction” we added: Therefore, the present study first investigated the effects of drug treatment and drug withdrawal on metabolic phenotypes in mice; Second, the effects of BPA on the expression of genes and proteins related to lipid metabolism in mouse liver under different treatment conditions were determined by Q-PCR and Western Blot(WB); Finally, molecular interference is used to validate the function of the relevant molecules in vitro. The aim is to identify potential molecular therapeutic targets for BPA-induced hepatic lipid metabolism disorders.

  1. The reference style should be corrected, it is currently incorrect in the text. Moreover, in line 81 the year of the citation is missing.

Response: Thank you very much for your comments. As suggested, we have corrected the reference style in the manuscript.

  1. It should be noted that the methodology should include catalog numbers of antibodies (and their dilutions), catalog numbers of primers (or their sequences if they were designed).

Response: Thank you very much for your comments. As suggested, we have added more details in the “Chemicals and Materials”. “Primary polyclonal antibody APOD (ab236868, Abcam, Cambridge, MA, USA, Dilution ratio:1/1000) and SCD1 (ab236868, Abcam, Cambridge, MA, USA). secondary antibody were purchased from Abcam (ab6721, Cambridge, MA, USA, Dilution ratio:1/1000).”

  1. Was the study approved by the ethics committee? The acceptance number should be provided.

Response: Thank you very much for your comments. As suggested, we have added the acceptance number of the ethics committee in the manuscript.

  1. The description of the results is correct, but it could be written how much exactly the change in the expression of genes and proteins is.

Response: Thank you very much for your comments. As suggested, we have rephrased the sentences in the manuscript.

  1. The discussion is satisfactory. Lines 308-309 should be deleted or authors should add conclusions.

Response: Thank you very much for your comments. As suggested, we have added the conclusions in the manuscript.

Thank you again for your comments. If you have any questions, please let me know as soon as possible.

Sincerely

Shaohua Yang or Hui-Li Wang

College of Food and Biological Engineering

Hefei University of Technology

193 Tunxi Road, Hefei 230009, Anhui, China

Tel/Fax: +86-551-65785485

Email: [email protected] or [email protected].

Round 2

Reviewer 1 Report

English needs to be corrected by an English-speaking person. A high number of sentences are gramatically incorrect.

Author Response

Dear editor:

Thank you very much for providing an opportunity for us to revise our paper. We also wish to thank you and reviewers for the positive comments and constructive suggestions on the manuscript, which greatly improve the manuscript. Accordingly, we have made the essential revisions suggested by the reviewers。

Thank you again for your comments. If you have any questions, please let me know as soon as possible.

Sincerely

Shaohua Yang or Hui-Li Wang

College of Food and Biological Engineering

Hefei University of Technology

193 Tunxi Road, Hefei 230009, Anhui, China

Tel/Fax: +86-551-65785485

Email: [email protected] or [email protected].

Reviewer 2 Report

The revised version manuscript is ready for publication.

Author Response

(The authors gave the same response as above.)
